# Natural scene sampling reveals reliable coarse-scale orientation tuning in human V1

Zvi N. Roth ●[1] ✉, Kendrick Kay[2,3] & Elisha P. Merriam ●[1,3]

Orientation selectivity in primate visual cortex is organized into cortical columns. Since cortical columns are at a finer spatial scale than the sampling resolution of standard BOLD fMRI measurements, analysis approaches have been proposed to peer past these spatial resolution limitations. It was recently found that these methods are predominantly sensitive to stimulus vignetting - a form of selectivity arising from an interaction of the oriented stimulus with the aperture edge. Beyond vignetting, it is not clear whether orientation-selective neural responses are detectable in BOLD measurements. Here, we leverage a dataset of visual cortical responses measured using high-field 7T fMRI. Fitting these responses using image-computable models, we compensate for vignetting and nonetheless find reliable tuning for orientation. Results further reveal a coarse-scale map of orientation preference that may constitute the neural basis for known perceptual anisotropies. These findings settle a long-standing debate in human neuroscience, and provide insights into functional organization principles of visual cortex.

Neurons in human visual cortex are organized according to their functional selectivity. A number of stimulus features are organized in coarse-scale cortical maps. For example, retinotopic visual cortex is organized as a log-transform of the visual field, with the polar dimensions of visual space (angle and eccentricity) corresponding to the Cartesian dimensions of the cortical surface[1–3]. Additionally, receptive field size[4,5] and preferred spatial frequency[6] increase monotonically with visual eccentricity. Selectivity to other visual features, such as ocular dominance and temporal frequency, are organized at a fine spatial scale, often forming columns through the cortical depth[7–10]. As in primates and carnivores[11], orientation selectivity in humans has been shown to also be organized at a fine spatial scale, in cortical columns that are approximately 0.7–0.8 mm in width across the cortical surface of primary visual cortex[12].

In addition to the fine-scale columnar architecture, fMRI studies have offered evidence for a coarse-scale organization of orientation preference[13–16]. Because of the relatively low spatial resolution of standard fMRI measurements (typically around 2 mm × 2 mm × 2 mm), these studies do not reveal the fine-scale columnar architecture. Rather, these studies have leveraged the broad spatial coverage afforded by fMRI to reveal a radial bias of orientation preference: voxels respond more strongly to orientations that point from the receptive field center toward fixation. This radial bias was originally assumed to reflect a physiological map of orientation selectivity[13], analogous to maps of receptive field location and spatial frequency selectivity. However, this radial bias map, it turns out, does not necessarily reflect solely a physiological map. Instead, it was shown to likely be, in large part, the result of stimulus vignetting, an interaction between the edges of the stimulus aperture and the spatial frequency envelope of the stimulus[17,18]. The principle underlying stimulus vignetting is that the Fourier spectrum changes in the vicinity of a change in contrast, such as an edge. In typical orientation mapping experiments, the assumption is that each condition contains a single orientation. But because of vignetting, this assumption isn't valid: different regions in the image contain different orientations and different Fourier power. Specifically, at the stimulus edge there is more power for the radial orientation than for other orientations. These observations challenge the interpretation of a large body of studies over the past 20 years that were presumed to measure orientation-selective responses in humans[19].

[1]Laboratory of Brain and Cognition, National Institute of Mental Health, NIH, Bethesda, MD, USA. [2]Center for Magnetic Resonance Research (CMRR), Department of Radiology, University of Minnesota, Minneapolis, MN, USA. [3]These authors jointly supervised this work: Kendrick Kay, Elisha P. Merriam. ✉e-mail: zvi.roth@nih.gov

If previous fMRI studies do not provide clear evidence regarding a physiological map of orientation selectivity, how is orientation actually represented in human visual cortex? Does the human brain contain a map for orientation selectivity at a coarse spatial scale, distinct from columnar architecture? Alternatively, is the presumed fine-scale columnar map the sole organizational principle for orientation in visual cortex? The answer to these questions has been obfuscated by stimulus vignetting: if there is indeed a coarse-scale map for orientation, it may be entirely eclipsed by stimulus vignetting.

To overcome these challenges, we apply a computational framework for studying orientation selectivity that explicitly models the effects of stimulus vignetting in order to access orientation-selective signals that would be otherwise obscured. To this end, we leverage a massive 7T fMRI dataset, the Natural Scenes Dataset (NSD), consisting of an extensive sampling of responses to natural scene stimuli in a small number of intensively-studied participants[20,21]. A large number of measurements and unique stimuli, combined with the high signal-to-noise ratio (SNR) of the fMRI measurements, enables us to robustly fit models that include dozens of parameters per voxel.

Image-computable models have been used to study a wide range of questions in visual neuroscience[17,22–24]. To assess orientation selectivity, our modeling approach exploits two image-computable models based on the steerable pyramid[25]. The constrained model, which includes both visual field position tuning and spatial frequency tuning, but pools equally across orientation-selective filters, is sensitive to the effects of stimulus vignetting[17]. This model is based on the model used previously to demonstrate stimulus vignetting[17], and it can fully explain responses modulated by total Fourier power. However, because the orientation filters at each level are pooled before the filter responses are computed, the model cannot capture any information about orientation. In other words, any apparent orientation selectivity in the model output is entirely due to stimulus vignetting. The full model, on the other hand, which allows unequal contributions from orientation-selective filters, is sensitive to orientation selectivity beyond the effects of stimulus vignetting (Fig. 1). Combined, the pair of models enables us to assess voxel-wise orientation selectivity while simultaneously accounting for the impact of stimulus vignetting. The modeling results reveal the existence of physiological maps for orientation preference.

## Results

### Both constrained and full models explain roughly similar amounts of variance

The NSD dataset contains measurements of 7T BOLD fMRI responses from 8 participants who each viewed 9000–10,000 distinct color natural scenes (22,000–30,000 trials) over the course of 30–40 scan sessions[26]. We fit two models characterizing V1 voxel responses to the natural scene stimuli. Both models fit voxel responses as a weighted sum of steerable pyramid filters. The constrained model pools across orientation, and is, therefore effectively composed of a range of spatial frequency filters across the visual field. In contrast, the full model includes flexible weights for both spatial frequency and orientation-tuned filters (Fig. 1). In order to evaluate how well each model fits the data, we cross-validated both models and measured the variance explained by each model on out-of-sample data.

The constrained and the full models explained a similar portion of the variance in the BOLD measurements (mean $R^2$: full 0.0311, constrained 0.0310) (Fig. 2A). If BOLD activity reflects orientation selectivity, we would expect the constrained model to be unable to account for modulations driven by local orientation differences between images. The full model, however, includes orientation tuning, and, therefore should be able to account for this additional response variability, assuming that the parameters that characterize orientation tuning can be reliably estimated. But we found that both models performed comparably, explaining similar amounts of variance, with the

constrained model slightly but significantly outperforming the full model (median $R^2$: full 0.0204, constrained 0.0212; two-sided Wilcoxon signed-rank test $p$ value $< 10^{-10}$). Moreover, cross-validated $R^2$ was highly correlated between the two models ($r = 0.9835$, $p < 10^{-30}$), likely reflecting gross differences in signal-to-noise ratio across voxels (Fig. 2B).

At first glance, these results suggest that there is no reliable orientation tuning to be modeled. However, since the full model has many more free parameters than the constrained model (57 vs. 8 parameters), we expect that, for voxels with low SNR, the full model would likely result in overfitting. For voxels with high SNR, on the other hand, the full model may be able to capture orientation tuning and lead to higher cross-validated $R^2$ values. Therefore, inspecting model performance only using summary statistics (e.g., median or mean) does not provide a complete picture, and it is necessary to analyze how model performance varies across voxels.

### Model performance depends on voxel SNR

For each individual voxel, model filter outputs were sampled using a population receptive field (pRF) estimated for that voxel from an independent pRF-mapping experiment that was conducted as part of the NSD dataset. In both the constrained and full models, the estimated pRF determined the portion of stimulus from which the model output is sampled (see Methods: *pRF sampling*). Hence, any error in estimating the size or location of the pRF would propagate forward, adversely affecting the model fits. Specifically, if the pRF estimate is inaccurate, the model would attempt to explain BOLD activity based on the portion of the natural scene image dictated by the pRF estimate, while the voxel would, in fact, be driven by another portion of the image. Therefore, it is likely that model performance depends on the quality of the pRF estimate.

Sorting voxels according to pRF $R^2$, we found that this was indeed the case. $R^2$ of both models were correlated with pRF $R^2$ (Constrained: $r = 0.5859$, $p < 10^{-30}$; Full: $r = 0.5728$, $p < 10^{-30}$). Our interpretation is that pRF $R^2$ is a good proxy for voxel-wise SNR, and that voxels with high SNR will tend to have both high-quality pRF estimates and good model performance on the natural scene responses (Fig. 2C, D). We furthermore found that the amount of additional variance that the full model explained beyond the constrained model was also correlated with voxel pRF $R^2$ ($r = 0.2418$, $p < 10^{-30}$) (Fig. 2E). This result implies that in cases of low SNR, including orientation tuning in an encoding model will likely result in overfitting, and that doing so is unlikely to reveal reliable orientation selectivity. However, for high SNR voxels in the NSD dataset used here, we are able to estimate reliable orientation selectivity that improves generalization on out-of-sample data. Additional analyses (Fig. 2G) indicate that voxels with high SNR tend to lie away from the fovea, and it is for these more peripheral voxels that we can reliably estimate orientation selectivity.

### Coarse-scale map of orientation selectivity

Explicitly modeling voxel responses enabled us to capture robust orientation selectivity that is not due to stimulus vignetting. What is the source of this orientation tuning? Although the voxel size in the NSD dataset (1.8 mm × 1.8 mm × 1.8 mm) is much larger than the size of orientation columns, fMRI studies using multivariate decoding methods have suggested that even with large fMRI voxels (3 mm × 3 mm × 3 mm), V1 voxel responses might exhibit robust orientation biases originating from a random sampling of cortical columns[27–29]. Therefore, we ask: is orientation preference scattered in a salt-and-pepper fashion, suggesting a random bias from orientation column sampling, or is orientation preference organized in a systematic map across the cortical surface?

To assess the potential organization of orientation selectivity, we plotted voxel orientation preference in visual space (Fig. 3A). We observe a clear coarse-scale map of orientation: preferred orientation

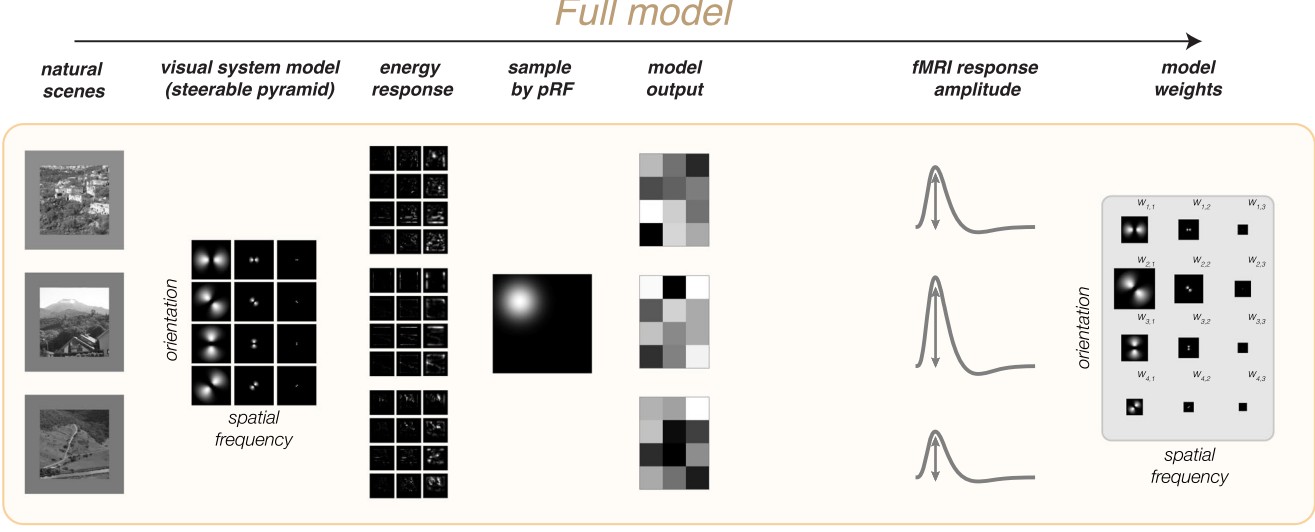

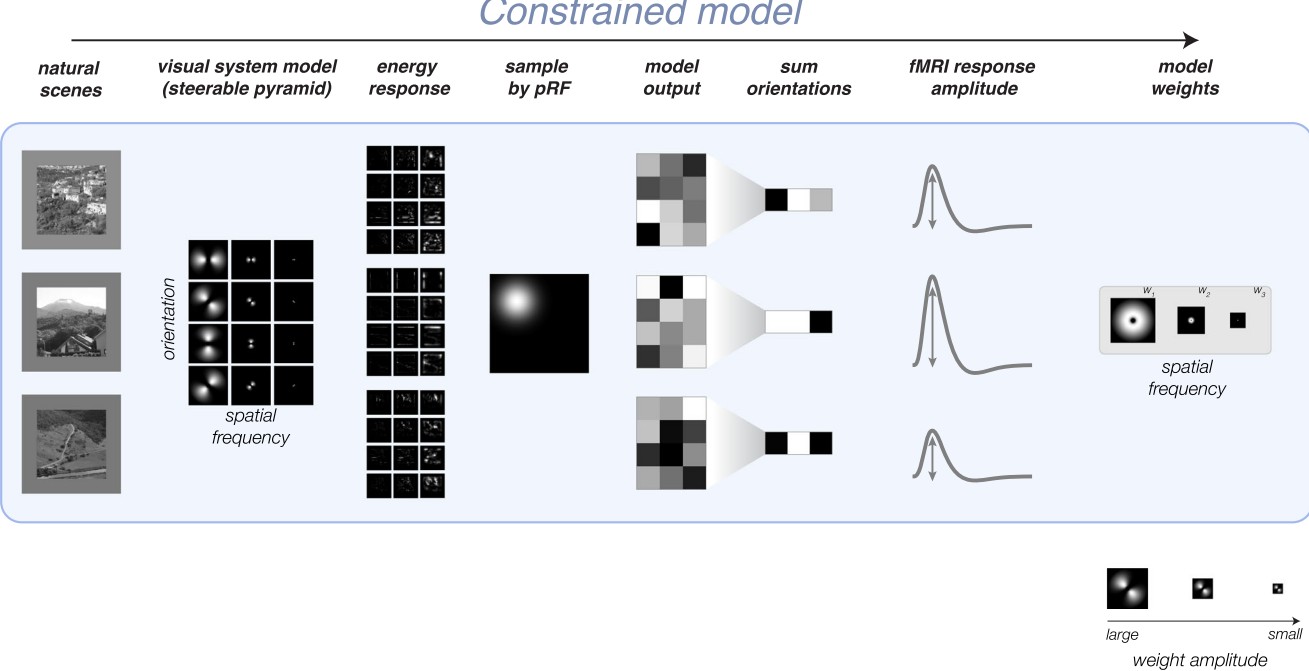

**Fig. 1 | Analysis pipeline for a single voxel.** Top, full model. Bottom, constrained model. Natural scene images were converted to grayscale and passed as input to a steerable pyramid (shown here with only four orientations and three spatial frequencies for visualization purposes). Each filter yielded an energy response that was sampled by the voxel pRF, resulting in a scalar output for each filter. Linear regression was performed on the response amplitudes observed for each voxel with filter output values as predictors. This procedure yielded a set of model weights, which were subsequently used to simulate responses to a range of gratings in order to determine the voxel's preferred orientation and spatial frequency. The constrained model (bottom) was identical to the full model except for an additional step of summing model outputs across orientations. As a result, the constrained model yielded a weight for each spatial frequency filter but enforced equal contribution across orientations. Example images shown here were created by the authors for illustration only and were not used in the study.

varies smoothly across the visual angle. This observation implies that voxels with nearby pRF locations have similar orientation preference, unlike the expectation based on the salt-and-pepper organization. Indeed, when we visualize cortical maps of estimated orientation preference, we see in V1 a clear progression of orientation that mimics the well-known organization of visual field angle (Fig. 3B).

The gradual progression of orientation preference across the visual field and the cortical surface provides evidence against a random-sampling bias of orientation columns. Instead, this observation reveals an organizational principle for visual cortex, namely, a coarse-scale spatial map for orientation preference.

### Radial map of orientation selectivity

The orientation map appears to resemble a radial map (Fig. 3A), but quantitative analysis is necessary to test the hypothesis that the map is indeed radial. We quantified the similarity of the orientation map to a radial map by computing the angular deviation of each voxel's orientation preference from the preference predicted by a perfectly radial map (Fig. 4, right). Deviation from radial as a function of pRF eccentricity resembled an inverted U (Fig. 5A): deviation was maximal at the fovea and at high (>5 deg) eccentricities, and lowest at intermediate (2–4 deg) eccentricities. We also compared the measured orientation map to two alternative possibilities: a vertical map and a cardinal map

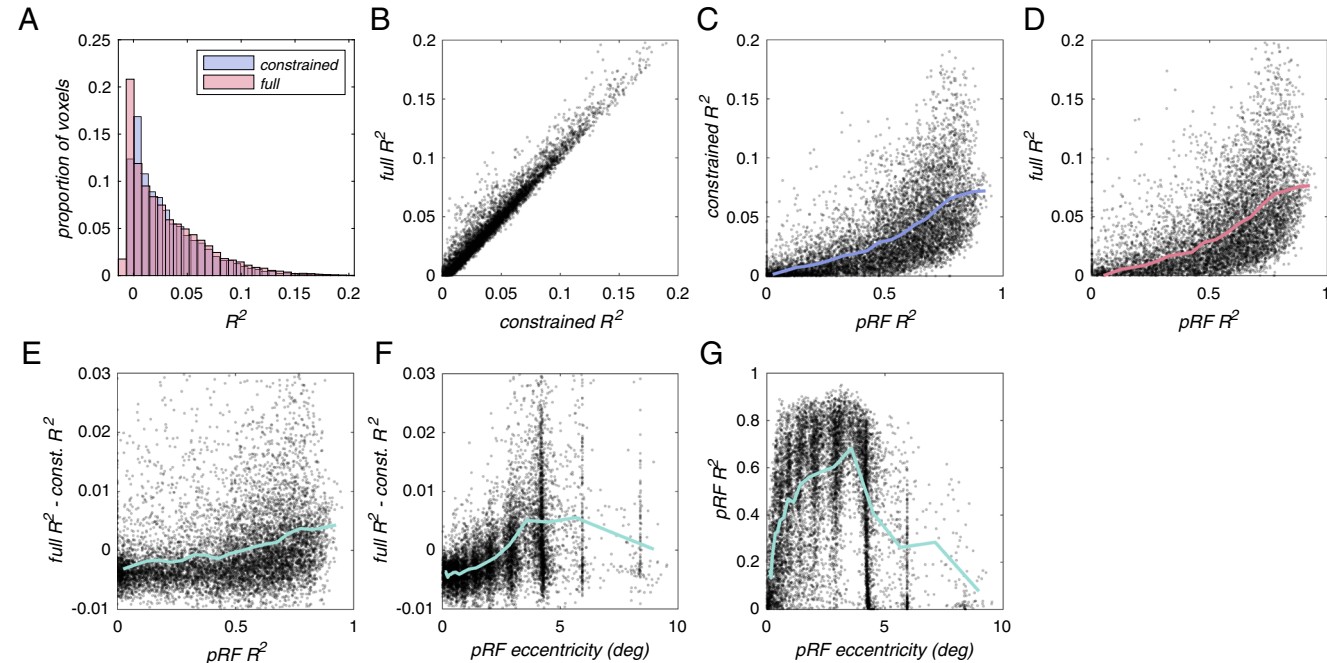

**Fig. 2 | Both constrained and full models fit voxel responses. A** Distribution of $R^2$ values for full (pink) and constrained (blue) models. Distributions for both models were similar, with the constrained model $R^2$ median slightly but significantly higher ($p = 7.4 \times 10^{-11}$, two-sided Wilcoxon rank-sum test). **B** $R^2$ values for the full model as a function of $R^2$ values for the constrained model. **C** $R^2$ values for constrained model as a function of pRF $R^2$. **D** $R^2$ values for the full model as a function of pRF $R^2$. **E** $R^2$ values for full model minus $R^2$ for a constrained model as a function of pRF $R^2$. **F** $R^2$ values for a full model minus $R^2$ for a constrained model as a function of pRF eccentricity. **G** pRF $R^2$ values as a function of pRF eccentricity. Solid lines in **C–G** indicate a running mean computed over 20 bins.

(Figs. 4, 5A). Deviation from the ideal radial map was lower than for either the ideal vertical or the ideal cardinal maps (Fig. 5B, C), indicating that orientation selectivity in V1 is approximately organized in a radial map.

Deviation from radial was not uniform across pRF polar angle (Fig. 5A, right), but was lowest at the horizontal and vertical meridians (Fig. 5A, orange), similar to the deviation from cardinal (Fig. 5A, green). Confirming this observation, deviation from radial was linearly correlated with angular distance from the closest meridian ($r = 0.141$, $p < 10^{-38}$). However, voxel-wise SNR is a potential limiting factor in this analysis, since voxel-wise SNR (as indexed by pRF $R^2$) correlates slightly negatively with distance from the meridians ($r = -0.055$, $p < 10^{-6}$). Nevertheless, when controlling for the effect of pRF $R^2$, the partial correlation between radial deviation and distance from the meridian remained strong ($r = 0.133$, $p < 10^{-34}$). We interpret this to mean that modulation of the radial bias by distance from the meridian is a feature of the coarse-scale orientation map, rather than a trivial result of anisotropies in pRF SNR across the visual field.

Ideal radial and vertical maps are identical at the vertical meridian, and differ maximally at the horizontal meridian (Fig. 4). Therefore, comparing the empirical orientation map with radial and vertical maps around the vertical meridian will not be particularly informative. Around the horizontal meridian, however, radial and vertical maps differ maximally. We, therefore, expected deviation from vertical to be roughly equal to deviation from radial around the vertical meridian. Around the horizontal meridian, on the other hand, we expected deviation from radial to differ maximally from deviation from vertical. Consistent with this expectation, deviation from radial was significantly lower than the deviation from vertical around the horizontal meridian, but not at the vertical meridian (Fig. 5B, right).

We found that radial and cardinal maps are identical at the horizontal and vertical meridians, and differ maximally at oblique (diagonal) angles (Fig. 4). Therefore, deviation from radial and cardinal should differ maximally around oblique angles and minimally around the meridians. Again, this was indeed the case: deviation from radial

was significantly lower than from cardinal only at oblique angles of the visual field (Fig. 5C, right).

Deviation from radial was significantly lower than the deviation from cardinal at intermediate eccentricities (Fig. 5C, left). This may be related to lower pRF $R^2$ around the fovea (Fig. 2G). Therefore, we cannot determine whether the radial bias is weakest at low eccentricities, or whether the larger deviation is entirely due to less accurate pRF estimates, while the strength of the radial map is, in fact, constant across eccentricities.

We conclude that the radial map is strongest around the meridians (i.e., at cardinal angles), and is weakest around oblique angles.

## Controlling for analysis pipeline

We have assumed thus far that orientation selectivity in the full model reflects variance that cannot be explained by the constrained model. We wondered if this result could somehow be an artifact of our modeling procedures, given the complexity of the model fitting pipeline. For example, it is conceivable that orientation selectivity in the full model (i.e., the weights the model assigns to different orientation filters), in fact, reflects variance that *can* be explained by the constrained model as well, but with other combinations of regressors. One example of this type of confound is stimulus vignetting[17], where apparent orientation selectivity, in fact, reflects spatial frequency tuning and not orientation tuning. Perhaps stimulus vignetting or other sources of response variance that do not genuinely reflect orientation tuning effectively masquerade as the orientation selectivity observed in the full model.

To test this possibility directly, we first regressed out all variance that was explained by the constrained model from the experimental data. We then fit the full model to these residual data. If the orientation selectivity of the full model reflects variance that can be explained by a combination of parameters in the constrained model, we would expect that regressing out that variance would leave the full model with no orientation-selective variance left to fit. If, on the other hand, orientation selectivity in the full model reflects only variance that cannot be

**Fig. 3 | Orientation preference changes smoothly across visual space and V1. A** Orientation preferences are plotted in visual space. Each line represents a single voxel and is positioned at the voxel's pRF center. Hue and orientation of the line indicate the preferred orientation. Line length, width, and scale reflect the amount of variance ($R^2$) explained by the constrained model. A solid square at ±4.2 deg indicates the size of the natural scene stimuli. See also individual subject plots in

Supplemental Fig. S1. **B** Orientation and pRF polar angle map overlaid on the left and right inflated "fsaverage" surfaces. For the angle map and unthresholded orientation map, all vertices in V1, V2, V3, and V4 are plotted. For the thresholded orientation map, only vertices with the top 50% full model $R^2$ are plotted. See also individual subject maps in Supplemental Fig. S2.

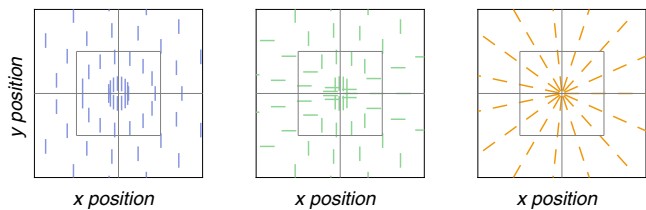

**Fig. 4 | Schematic of three ideal orientation maps.** Ideal maps of orientation preference are plotted in visual space. Each line represents a single ideal voxel and is positioned at the voxel's pRF center. The orientation of the line indicates the preferred orientation. Left, vertical map. Center, cardinal map. Right, radial map.

explained by the constrained model (i.e., true orientation tuning), then regressing out variance explained by the constrained model should have no effect on the full model orientation selectivity.

Consistent with the second scenario, the map derived from analyzing the residuals (Fig. 6A, B, left) was nearly identical to the original map (Fig. 3). Complementarily, fitting the full model to the variance

explained by the constrained model (i.e., the output predicted by the constrained model) resulted in a random map of orientation preference (Fig. 6A, B, right). This indicates that the orientation selectivity in the full model cannot be explained by the constrained model.

## Discussion
### Summary
By analyzing a unique, extensive visual fMRI dataset using a model-based framework, we obtained robust estimates of orientation selectivity in human visual cortex. Unlike previous fMRI studies in which measurements of orientation selectivity were potentially confounded by interactions between the oriented stimuli and the stimulus aperture, the orientation selectivity that we report here cannot be attributed to stimulus vignetting. We uncovered a radial bias of orientation selectivity that is coarse-scale and widespread throughout the extent of V1. The radial bias that we describe here is distinct from the spatial pattern expected by the sampling of cortical columns. This coarse-scale orientation bias may be a fundamental organizational principle of human V1, providing a physiological basis for well-documented behavioral biases in orientation judgements.

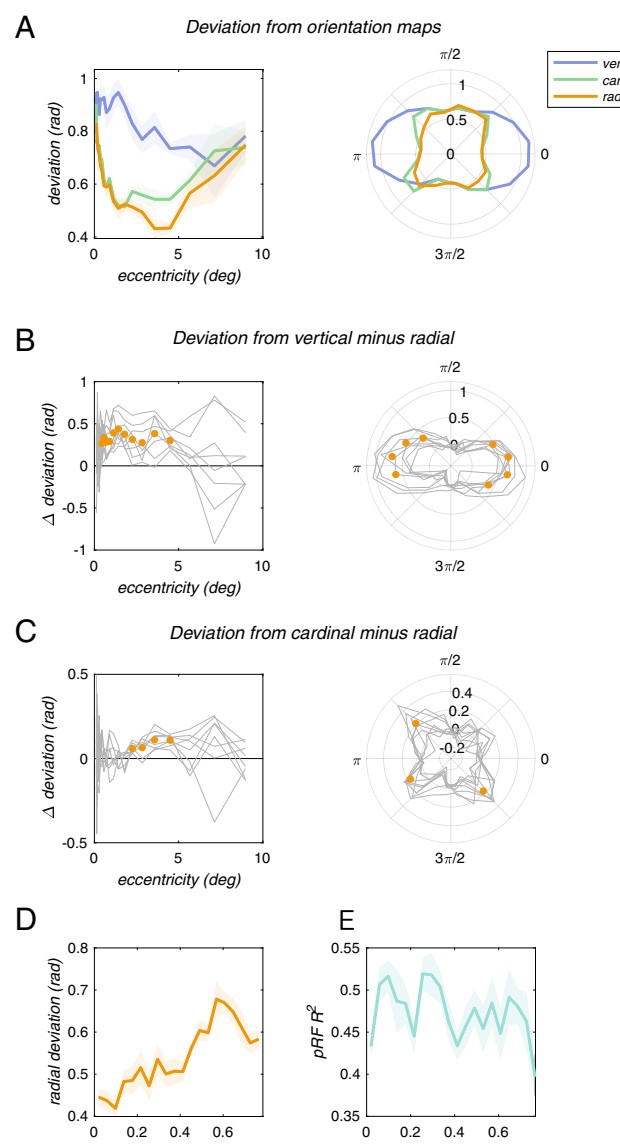

**Fig. 5 | Coarse-scale bias is mostly radial in organization.** **A** Deviation of preferred orientation from ideal vertical (blue), cardinal (green), and radial (orange) maps, as a function of pRF eccentricity (left) and pRF angle (right). **B** Deviation from vertical minus deviation from radial, as a function of pRF eccentricity (left) and pRF angle (right). Gray lines, individual subjects. Orange points, significantly lower deviation from radial than from vertical ($p < 0.05$, one-way $t$-test, seven degrees of freedom, no correction for multiple comparisons). **C** Deviation from cardinal minus deviation from radial as a function of pRF eccentricity (left) and pRF angle (right). Gray lines, individual subjects. Orange points, significantly lower deviation from radial than from cardinal ($p < 0.05$, one-way $t$-test, seven degrees of freedom, no correction for multiple comparisons). **D** Deviation from radial as a function of angular distance from the closest meridian. **E** pRF $R^2$ as a function of angular distance from the meridian.

## Multiple scales of stimulus representation in the human visual cortex

The human brain likely contains multiple representations of stimulus orientation at different spatial scales, co-existing within the same cortical visual area but arising from distinct neural computations. The first, and most familiar, is the fine-scale, columnar organization. The orientations of visual features are represented in an orderly pinwheel-like progression within each hypercolumn across the cortical surface[30–32]. While orientation columns have not been measured directly in humans using electrophysiological methods, based

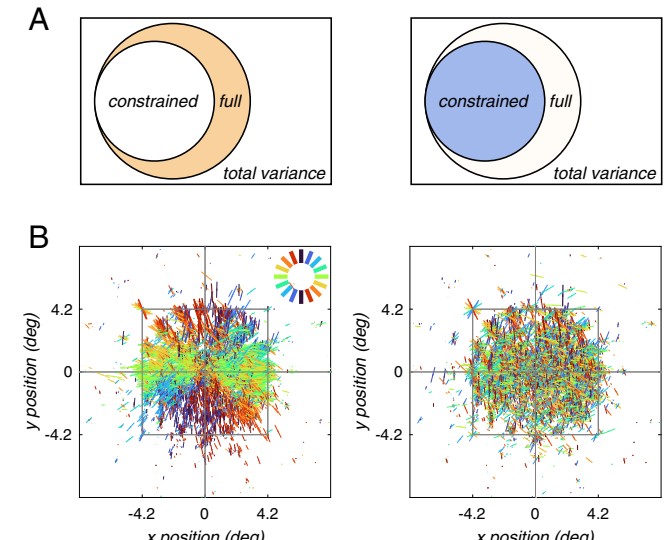

**Fig. 6 | Control analyses demonstrate that the full model indeed captures orientation selectivity that cannot be derived from the constrained model.** **A** Schematic Venn diagram illustrating the control analyses. First, the constrained model was fit. The full model was then fit either to the residual (left) or the output (right) of the constrained model. **B** Orientation map in visual space derived from the two analyses. Same format as Fig. 3A. The orientation map on the left is nearly identical to the map estimated from the original data (see Fig. 3A), indicating that the full model capitalizes on variance not modeled by the constrained model. In contrast, the orientation map on the right has no discernable organization, demonstrating that the constrained model cannot explain variance reflecting orientation selectivity.

on postmortem measurements of ocular dominance columns in humans[33,34], orientation columns are likely to be less than a millimeter wide along the cortical surface. In addition to this fine-scale columnar structure, a number of fMRI studies have also reported a second, coarse-scale map-like organization for orientation selectivity that is at the scale of the retinotopic organization of V1[13,14], spanning tens of centimeters, orders of magnitude larger than cortical columns. For the purposes of this discussion, we define a pattern of cortical activity to be "fine-scale" if it has features that are smaller than the point-spread function of a conventional fMRI voxel (~2 mm × 2 mm × 2 mm). According to this definition, the radial bias that we report here is clearly coarse-scale.

The presence of both fine- and coarse-scale patterns of orientation selectivity raises two fundamental questions. The first regards the scale of information leveraged by multi-voxel pattern analysis (MVPA) methods to decode orientation. In a landmark study, ref. [27] demonstrated that it is possible to use a linear classifier to decode the orientation of a grating presented to the subject on an individual trial. Cortical columns are irregularly organized with respect to the rectilinear voxel grid, which, it was posited, could lead to small biases in voxel responses that are decodable with MVPA. The conjecture that MVPA methods are sensitive to fine-scale signals had a profound impact on fMRI research, well beyond visual neuroscience, because it implied the feasibility of studying neural representations in the human brain that are instantiated at a spatial scale smaller than an fMRI voxel[28]. However, it has been surprisingly difficult to fully support or fully refute this claim, especially in light of an alternative account suggesting that MVPA is primarily sensitive to the coarse-scale orientation bias[13]. This uncertainty has engendered ongoing and unresolved debate[35–44]. The current study does not directly bear on this debate. Here, we quantified orientation information in fMRI BOLD responses using a neurally-inspired image-computable model. While our results reaffirm the existence of a prominent coarse-scale organization for

orientation (i.e., a radial bias), we did not use MVPA, nor did we explicitly model the scale of orientation information with a spatial model of the cortical surface. One fruitful approach may be to build spatial models of V1[3,45,46], which could provide a computational platform for testing hypotheses regarding the spatial scale of information and explicitly test the sensitivity of MVPA methods to different spatial scales of orientation information.

The second question concerns the neural response properties that give rise to the coarse-scale orientation bias. While the cortical architecture giving rise to fine-scale orientation columns is relatively well understood[47–49], much less is known about the origin of the coarse-scale bias. It was initially presumed that the coarse-scale bias stems from a pattern of cortical organization in which neurons with similar patterns of orientation selectivity form gradients over large swaths of cortex[13]. This idea was challenged by the insightful work of Carlson[18], who demonstrated through simulation that the radial bias could arise from properties of the stimulus aperture or "vignette", rather than from the pattern of orientation selectivity in the brain. This proposal suggested that stimuli with the same underlying orientation could be shown through different vignettes and this could reverse the observed orientation selectivity. These predictions have been confirmed empirically[17], suggesting that orientation selectivity measured with fMRI is indeed influenced to some degree by stimulus vignetting.

Is stimulus vignetting the primary driver of the coarse-scale orientation biases reported in previous fMRI studies? The magnitude of stimulus vignetting reported in ref. 17 was roughly commensurate with the magnitude of the radial bias reported in previous studies of orientation selectivity[13], suggesting that the radial bias could, in theory, be due to stimulus vignetting. fMRI studies have reported a radial bias using a wide range of stimulus parameters, i.e., different spatial frequencies and aperture sizes. Further theoretical simulations (Supplementary Fig. 4) suggest that each of these configurations could produce stimulus vignetting, depending on the location of the edge in the retinotopic map, the spatial frequency channels that contribute most to the voxel's response, and the form of response normalization assumed. On the other hand, there have been reports of orientation decoding away from the stimulus edge, suggesting orientation information in fMRI measurements that are not due to vignetting (ref. 27, Supplementary Fig. 3; ref. 41). These considerations have, up to now, left the field at an impasse, with no clear way to characterize true coarse-scale orientation selectivity in the face of a potential stimulus confound. Here, we turned to natural scene stimuli and an image-computable modeling approach to overcome these challenges. By explicitly modeling the spatial frequency and orientation content of each image, we account for the presence of a stimulus aperture, enabling the accurate characterization of orientation selectivity in the human visual cortex.

The characterization of a coarse-scale orientation bias does not preclude a contribution from cortical columns. It is likely the case that orientation-selective signals measured with fMRI are multi-scale[38,42], reflecting a contribution of both fine- and coarse-scale orientation signals, with the relative contribution dependent on critical experimental parameters, such as fMRI voxel size, acquisition method (gradient echo, spin echo, or VASO[50]), proximity of a voxel to veins[51,52], as well the stimulus protocol itself[53]. These considerations imply that various reports of orientation selectivity over the years may have differentially emphasized fine- and/or coarse-scale components of the signal depending on which combination of these parameters were used. Further work, perhaps using an extension of the image-computable model that we have developed here, may be able to tease apart the relative contribution of each of these orientation-selective signals.

### Multiple coarse-scale biases

Here we report a radial bias of orientation preference in visual cortex, extending a number of prior BOLD fMRI studies[13–16,54]. But while the radial bias in previous fMRI studies may have been entirely a result of

stimulus vignetting, the radial bias we report here is not explainable by the constrained model (Fig. 6), and therefore reflects a coarse-scale orientation map distinct from stimulus vignetting. In addition to a radial bias that was most pronounced in the periphery, a previous study identified a vertical bias at mid-eccentricities, closer to the fovea[16]. We tested for such a vertical bias, but found no evidence to support this possibility. At intermediate eccentricities, orientation preference was significantly closer to radial than to vertical, although close to fixation, the radial preference was not significant (Fig. 5B). This may be a result of lower SNR around fixation, as reflected in the lower pRF $R^2$ values (Fig. 2G). Therefore, we cannot definitively rule out a vertical bias at fixation, as reported by ref. 16.

A cardinal bias has been identified previously in the human visual cortex[55], in other primates[56–58], and in carnivores such as ferrets[59–61] and cats[62–65]. The cardinal bias is typically described as a stronger response to vertical and horizontal orientations compared to oblique orientations. To enable a comparison to the radial map, we defined a cardinal map as a stronger response to the meridian closest to the pRF center, which is radial only for pRFs along the meridian (Fig. 4, center). This definition describes an orientation map, relating orientation preference to retinotopic preference. The cardinal map, as we defined it, entails a cardinal bias, since, when averaging across all of V1[55], cardinal orientations evoke the strongest responses.

Although the orientation map was closer to a radial map than to the cardinal map, we did find that the radial bias was strongest around the vertical and horizontal meridians. This pattern can be described as cardinal modulation of a radial map, and is consistent with previous fMRI findings[15].

What physiological factors underly the cardinal modulation of the radial map? It is possible that the radial bias is stronger around the meridians because of stronger SNR in those regions, or because of small artifacts away from the meridians caused by cortical unfolding and veins[66,67]. However, we believe such artifacts and SNR differences should manifest similarly in the pRF data, yet we found that the lower pRF $R^2$ (Fig. 5E) could not fully explain the weaker radial bias away from the meridians. Similarly, a fine-scale columnar bias could potentially cause voxel preference to deviate from the radial orientation, but we would expect such an effect to take place uniformly across all polar angles. Instead, we believe it is more likely that the cardinal modulation reflects the true nature of the radial bias: it is possible there are two co-existing biases, a cardinal and a radial, or that neurons around the meridians show a stronger preference for the radial orientation.

### Source of coarse-scale orientation bias

How does the coarse-scale orientation map form? The mechanism is likely related to the source of orientation selectivity itself, which is still debated. When orientation-selective neurons were first discovered in the cat visual cortex, their tuning properties were proposed to arise from the convergence of center-surround neurons in the lateral geniculate nucleus (LGN) that were themselves not orientation-selective[68]. Local interactions between V1 neurons have also been shown to amplify orientation selectivity[69], suggesting that orientation selectivity arises from both the convergence of feedforward input and local circuit interactions. But it has also been suggested that orientation selectivity is computed earlier in the visual pathway, and that it is to some degree inherited by V1 neurons. Multiple lines of evidence point towards orientation selectivity being present already in some LGN neurons[70–75] and even in retinal ganglion cells[76–80], raising the possibility that orientation selectivity in V1 reflects computations at earlier processing stages. Consistent with this possibility, the retinal size relative to V1 size predicts across species whether orientation preference will be arranged in cortical columns or scattered in a salt-and-pepper fashion[81]. The coarse-scale orientation map may be the result of the same mechanisms that form orientation selectivity, or it may involve other unique factors. Determining the source of the coarse-

scale orientation bias, and whether it differs from sources of fine-scale selectivity, will require additional research involving measurements at a range of spatial scales.

A number of distinct neurophysiological mechanisms could, in theory, give rise to the coarse-scale bias uncovered here. For example, the radial bias could reflect a higher number of neurons preferring the radial orientation. Alternatively, the radial bias could reflect a higher firing rate for neurons preferring the radial orientation. A third possibility is that neurons preferring the radial orientation could have a narrower tuning bandwidth. All three of these scenarios presume a higher mean population firing rate in response to radial stimulus orientations, which would presumably translate to larger BOLD fMRI responses[82–85]. Since BOLD fMRI measures a hemodynamic signal and is an indirect measure of neural activity, it is also possible that the orientation tuning we measured here reflects synaptic inputs from either feedforward or feedback projections, or local field potentials[86,87]. Future electrophysiology studies in humans and non-human primates may shed more light on the relationship between the coarse-scale orientation bias and the underlying neurophysiology.

An alternate possibility is that coarse-scale orientation biases are an emergent property of visual cortex. When the visual system is modeled with large, unconstrained models, certain anisotropies and biases emerge, including a cardinal bias[88] and a radial bias[89,90]. These biases are due to statistics of the images used to train the models. Image statistics may underlie orientation biases in the human visual cortex as well[91,92], although it is unclear whether biases evident in neural networks account for similar biases we have observed in the human visual cortex.

### Natural scenes vs. oriented gratings

Most prior fMRI studies of orientation selectivity have relied on oriented gratings. This approach stems from a long and successful history in visual neurophysiology dating back to ref. 93. Such "synthetic" grating stimuli are optimal for driving individual V1 neurons because they can be presented in full-contrast and because the parameters of the stimuli (size, position, and spatial frequency) can be carefully tailored to the individual neuron being recorded. However, such gratings may be less appropriate when studying large neural populations, as with fMRI, since a single voxel reflects the pooled activity of many neurons with a wide range of selectivities, and therefore no single grating will be optimal for every neuron contributing to the voxel's response[19]. Natural scene stimuli are inherently broadband along multiple dimensions, and hence may be more appropriate for studying population responses. However, natural scene stimuli do have, on average, lower contrast than gratings, and will not drive individual V1 neurons at their maximal firing rates. Thus, natural scenes might not be the most efficient set of stimuli for estimating the voxel-wise encoding models used in the current study. Nonetheless, this loss of efficiency is counteracted by the massive number of trials in the dataset.

### Image-computable models and V1

A previous study[94] decoded natural images from BOLD responses in visual cortex by fitting encoding models to individual voxels. One finding was that including orientation tuning did not improve decoding beyond the accuracy obtained with only spatial frequency tuning (ref. 94, Supplemental Fig. 8). This result seems to be at odds with our findings here. However, a direct comparison between these studies is somewhat complicated. First, in a decoding approach, performance is sensitive to responses across multiple voxels and how they jointly encode stimuli. Such an approach yields results that are more difficult to interpret compared to a more straightforward approach in which the properties of individual voxels are examined. Second, although modeling orientation tuning in addition to spatial frequency tuning did not lead to improved decoding accuracy, when the spatial

frequency was absent or assumed to be identical across the entire region of interest, including orientation tuning did improve decoding. Thus, the results of the present study are not necessarily inconsistent with the previous study. Finally, it is important to note that the estimation of model parameters for the Gabor wavelet encoding model used in the previous study was performed using gradient descent with early stopping. This type of regularization (early stopping) reduces variance at the expense of introducing bias, and the exact nature of this bias is dependent, in a complex way, on the statistics of the model inputs (e.g., orientation statistics in natural images). A major advantage of the approach used in the present study is the use of unregularized ordinary least-squares for parameter estimation, which was made possible by the combination of the sheer size of the NSD dataset and the pRF constraints incorporated into our models. This approach avoids complications associated with regularization and facilitates accurate interpretation of voxel selectivity.

The image-computable model that we used here was based on the steerable pyramid[25], a sub-band image transform that decomposes an image into orientation and spatial frequency channels (see Methods: Steerable Pyramid). In our previous study of stimulus vignetting[17], we made two simplifying assumptions. First, because most of the power in the stimulus was at a single spatial frequency, we only analyzed the response of the model at a single spatial frequency channel centered at the spatial frequency of the stimulus (Fig. S4). This approach provided good qualitative fits to the fMRI data, and we further confirmed that an alternative approach of averaging across all the channels did not change the model predictions (Fig. S4). In the present study, such an approach is not feasible since the naturalistic images are broadband in spatial frequency. Instead, our modeling approach enabled fitting weights to all model channels, essentially estimating a spatial frequency tuning curve for each voxel. An alternative approach would be to weigh the different channels according to independent measures of spatial frequency tuning for each voxel[6,95]. The Natural Scenes Dataset could then be used to fit only the weights on orientation channels, which may result in more accurate estimates of orientation preference because of the smaller number of free parameters.

The second simplifying assumption involves scaling the channel outputs. For each orientation and spatial frequency, the pyramid includes a quadrature pair: two RFs with different phases. The sum of the squares of the responses of the two RFs is typically taken, yielding an "energy" response, which uniformly tiles all orientations and spatial frequencies[96,97]. This energy response is often nonlinearly scaled in order to better match the contrast-response function of V1 neurons. However, determining the model architecture and normalization pool appropriate for an fMRI voxel is not trivial and is very much an area of active investigation[98–101]. We acknowledge that the form of scaling could have an impact on the size and spatial extent of vignetting. Determining the most appropriate scaling is an important issue that remains unresolved. In the context of the current study, we think it unlikely that the main results are sensitive to the particular regime of scaling employed.

### Behavioral correlates of the coarse-scale bias

Understanding the stimulus selectivity of neurons and their organization is fundamental for understanding how neural computations lead to visual perception. In particular, coarse-scale organizations are likely critical elements of neural accounts of behavior, since large populations of neurons are likely to contribute to the final behavioral readout.

A behavioral radial bias has been reported by several psychophysics studies: sensitivity is higher to radial orientations than to other orientations[14,102–104]. Another well-known bias is the oblique effect: across the visual field, sensitivity is higher to cardinal orientations (vertical or horizontal) than to oblique orientations (diagonal)[103–107]. The physiological orientation selectivity measured here may underlie both behavioral effects. The radial map may be the source of the

 

behavioral radial bias, while the cardinal modulation may underlie the behavioral oblique effect. It has been suggested that fMRI response amplitudes reflect the neural SNR, which in turn determines the perceptual performance[108]. In that case, a stronger fMRI response to the radial orientation should correspond to higher perceptual performance for stimuli with radial orientations.

A major endeavor in neuroscience is to link brain properties with behavioral readout, and visual neuroscience has made significant progress toward this goal. Recently it has been shown that V1 size and cortical magnification in individual subjects is correlated with contrast sensitivity[109]. Similarly, the extent of cortical magnification in V1 corresponds to orientation discrimination performance in individual participants[110]. If the coarse-scale map revealed here constitutes the neural basis for behavioral anisotropies, we hypothesize that individual differences in the orientation map that we report here are related to individual differences in perception. Successfully demonstrating such a correspondence would provide a crucial link between brain and behavior.

## Methods

### Natural scenes dataset
The fMRI data analyzed here is from the Natural Scenes Dataset (NSD; http://naturalscenesdataset.org)[20]. The NSD dataset contains measurements of fMRI responses from eight participants who each viewed 9000–10,000 distinct color natural scenes (22,000–30,000 trials) over the course of 30–40 scan sessions. Scanning was conducted at 7 T using whole-brain gradient-echo EPI at 1.8-mm resolution and 1.6-s repetition time. Images were taken from the Microsoft Common Objects in Context (COCO) database[111], square cropped, and presented at a size of 8.4° × 8.4°. A special set of 1000 images were shared across subjects; the remaining images were mutually exclusive across subjects. Images were presented for 3 s with 1-s gaps in between images. Subjects fixated centrally and performed a long-term continuous recognition task on the images. The fMRI data were pre-processed by performing one temporal interpolation (to correct for slice time differences) and one spatial interpolation (to correct for head motion). A general linear model was then used to estimate single-trial beta weights. Cortical surface reconstructions were generated using FreeSurfer, and both volume- and surface-based versions of the beta weights were created.

In this study, we used the 1.8-mm volume preparation of the NSD data and version 3 of the NSD single-trial betas in percent signal change units (betas_fithrf_GLMdenoise_RR). The results in this study are based on data from all NSD scan sessions, from all eight subjects who participated in the NSD study.

### Stimuli
NSD images were originally 425 × 425 pixels, and were then upsampled for display purposes to 714 × 714 pixels. We reproduced this upsampling in our stimulus preparation, and padded the images with a gray border on all four sides (mimicking the scanner display environment), resulting in a final image dimension of 1024 × 1024 pixels (12.05° × 12.05°). A semitransparent red fixation point was added at the center to simulate the actual stimulation experienced by the subjects during the experiment. Images were converted to grayscale by averaging across the three color channels. To speed up subsequent computations, the images were then downsampled to 512 × 512 pixels. To enable cross-validation, the set of 10,000 images assigned to each subject was randomly divided into two partitions of 5000 images each. For subjects who completed fewer than 40 sessions, only the viewed images were used, which resulted in a slightly different number of images included in each partition.

### Steerable pyramid
We built two models based on the steerable pyramid[25]: a full model and a constrained model. The full model simulates each neuron in V1 with a receptive field that is tuned for both spatial frequency and orientation, and then allows for variable weighting of these model neurons. The constrained model also simulates populations of V1 neurons, but enforces equal weighting of model neurons across orientation by summing across orientation subbands of the pyramid. It is possible to create steerable pyramid models with a wide range of parameters, each instantiating different hypotheses regarding the tuning properties of individual neurons.

We used a steerable pyramid with eight orientations, seven spatial frequency levels, and a spatial frequency bandwidth of one octave, resulting in tuning profiles that resemble those of individual V1 neurons[112]. The pyramid, and the full model, had a total of 56 filters. After summing across the eight orientation filters, the constrained model consisted of seven filters. The number of spatial frequency levels was determined by the size of the image (512 × 512) and the spatial frequency tuning bandwidth (1 octave). This results in seven filters, with preferred spatial frequencies of 128, 64, 32, 16, 8, 4, and 2 cycles per image. These values were then converted to cycles per degree given the size of the image in degrees (12.05°): 21.85, 10.93, 5.46, 2.73, 1.37, 0.68, and 0.34 cycles/degree (cpd) (Fig. S3). In our previous work with the steerable pyramid, we used only the level corresponding to the stimulus spatial frequency. In this study, all levels were fit to the data. We chose to have eight orientations, two more than in our previous study, in order to increase the accuracy of the estimated orientation preference, while maintaining tuning width that was comparable to those measured in primate electrophysiological recordings[112]. For each orientation and spatial frequency, the pyramid includes a quadrature pair: two RFs with different phases. We take the sum of the squares of the responses of the two RFs, yielding an "energy" response, which uniformly tiles all orientations and spatial frequencies[96,97].

### pRF modeling
pRF estimates are included in the NSD, where full details are found[20]. Briefly, pRFs were estimated based on a single session (six runs, 300 s each) of a pRF-mapping experiment. Stimuli consisted of slowly moving apertures (bars, wedges, and rings) filled with a dynamic colorful texture, that appeared within a circular region of 8.4 deg diameter. Subjects performed a color change detection task at fixation. pRFs were estimated using the Compressive Spatial Summation (CSS) model[100].

### Regions of interest
Regions of interest V1, V2, V3, and hV4 were defined in the NSD dataset based on the pRF maps. In this study, we analyzed all four regions but focused on V1, where orientation selectivity has been studied extensively. Results are presented for V1 only, except for the surface maps (Fig. 3 and Supplemental Fig. 2), which show all regions.

### pRF sampling
The output of each filter in the steerable pyramid was sampled by each voxel's pRF by multiplying the 2D pRF with the filter output. The pRF was modeled as a 2D isotropic (circular) Gaussian, using the "size" parameter as the Gaussian's standard deviation. (Note that the "size" parameter, as estimated as part of NSD, reflects the response of the modeled pRF to point stimuli and takes into account the exponent used in the CSS model.) For filter k of image j ($F^{j,k}$), the sampled output for voxel $i$ with a pRF centered at $(x_i, y_i)$ and standard deviation of $\sigma_i$, is computed as the dot product between the pRF and the filter:

$$f_i^{j,k} = \sum_{x,y} F_{x,y}^{j,k} \cdot e^{-\frac{(x_i - x)^2 + (y_i - y)^2}{2\sigma_i^2}} \tag{1}$$

The full model had 56 sampled outputs per image, for each voxel. For the constrained model, sampled outputs were summed across

orientations. Thus, the constrained model had seven sampled outputs per image, for each voxel.

## Multiple regression

We modeled the responses of voxel $i$, $y_i$, as a linear combination of the sampled filter outputs plus noise:

$$y_i = f_i \bullet \beta_i + \varepsilon_i \tag{2}$$

Here $f_i$ is a matrix consisting of voxel $i$'s sampled outputs for all filters of all images and a constant term (images × filters + 1). $\beta_i$ is a vector of beta weights (filters + 1 × 1), and $\varepsilon_i$ is a set of residuals (images × 1).

Beta weights were estimated using ordinary least-squares:

$$\hat{\beta}_i = (f_i^T f_i)^{-1} f_i^T y_i \tag{3}$$

Note that each voxel not only had different beta weights but also different predictors due to the incorporation of each voxel's unique pRF, thus distinguishing this regression from a general linear model analysis of the voxel responses.

To assess model accuracy, we performed cross-validation. After estimating model parameters on one-half of the data, the regression prediction was calculated as:

$$\widetilde{y}_i^{\text{pred}} = f_i \bullet \widetilde{\beta}_i \tag{4}$$

where $f_i$ is constructed for the other half of the data, and $\widetilde{\beta}_i$ are the betas weights estimated using the other partition. The residual of this prediction is given by

$$\widetilde{y}_i^{\text{resid}} = y_i - \widetilde{y}_i^{\text{pred}} = y_i - f_i \bullet \widetilde{\beta}_i \tag{5}$$

Cross-validated $R^2$ is then computed as

$$R_i^2 = 1 - \frac{SS(\widetilde{y}_i^{\text{resid}})}{SS(y_i - \bar{y}_i)} \tag{6}$$

where $\bar{y}_i$ is the mean response across images, and SS denotes the sum of squares.

Regression was performed separately for the full model and for the constrained model on each of the two partitions. Regression coefficients and $R^2$ values were then averaged across partitions.

## Inferring preferred orientation and spatial frequency

After estimating the optimal weights for each voxel, we simulated an electrophysiology experiment for quantifying neural tuning, by probing the model with gratings at different orientations and spatial frequencies and measuring its predicted response. Full contrast gratings were 512 × 512 pixels at 30 spatial frequencies ranging from a single (horizontal) cycle in the image (0.083) to the Nyquist frequency (21.25 cpd), spaced exponentially. For each spatial frequency, gratings were oriented at 30 different angles, spaced uniformly between 0 and pi. All gratings were then passed through the steerable pyramid, and each filter's outputs were summed. Voxel responses to the gratings were simulated by multiplying the model outputs for each grating with the voxel's filter weights. The preferred spatial frequency of a voxel was estimated by first averaging simulated responses across orientations and then computing the mean frequency, weighted by response amplitudes. Similarly, the preferred orientation of a voxel (for the full model) was estimated by averaging across frequencies and then computing the circular mean, weighted by response amplitudes. For all weighted means, the minimal weight was first subtracted from all weights to eliminate any negative weights.

## Cortical surface maps

In order to create a group map on the cortical surface, each subject's data in volume space was transformed to surface space using nearest-neighbor interpolation, using the mrTools toolbox in Matlab[113]: each vertex was assigned the value of a single voxel, and multiple vertices could inherit values from the same voxel. Next, all subjects' surface data were transformed to a single cortical space, FreeSurfer's "fsaverage" space. For the group map, we computed the circular mean across subjects for each vertex in V1–V4, weighted by the full model $R^2$ values. The resulting map of mean orientation preference was displayed on the "fsaverage" inflated cortical surface (Fig. 3B).

## Quantifying coarse-scale biases

We quantified the strength of the coarse-scale orientation map by comparing it to an ideal, perfectly radial map, as well as ideal vertical and cardinal maps (see Fig. 4). For each voxel, we computed the circular angular distance from the preferred orientation to the predicted orientation. For the radial map, the predicted orientation was the radial orientation, according to the voxel's pRF angle (Fig. 4, right). For the cardinal map, the predicted orientation was vertical for pRF angles closer to the vertical meridian than to the horizontal meridian, and horizontal for pRF angles closer to the horizontal meridian than to the vertical meridian (Fig. 4, center). For the vertical map, the predicted orientation was vertical for all voxels (Fig. 4, left). To average across voxels, we divided voxels into 20 bins according to voxel pRF eccentricity, pRF angle, and pRF $R^2$. For eccentricity binning, bin width increased exponentially with eccentricity. Voxels with pRF $R^2$ values below zero were excluded from binning. To compare the strength of the radial bias and other alternative biases, we averaged across voxels within each bin separately for each subject, and subject means were then submitted to a paired-sample $t$-test, with seven degrees of freedom.

## Control analysis: analyzing regression residuals

For this analysis, after performing multiple regression with the constrained model as predictors, we took the residuals from the same partition used to estimate the beta weights. We then performed regression on the residuals, this time using the full model.

## Control analysis: analyzing regression prediction

For this analysis, after performing multiple regression with the constrained model as predictors, we multiplied the regression coefficients of the same partition with the predictors to get the regression prediction. We then performed regression on the prediction, using the full model as predictors.

## Reporting summary

Further information on research design is available in the Nature Research Reporting Summary linked to this article.

## Data availability

The NSD dataset is freely available at http://naturalscenesdataset.org. Images used for NSD were taken from the Common Objects in Context database (https://cocodataset.org). Source data are provided with this paper.

## Code availability

Code for analyzing the data and generating the figures is available at: https://github.com/elimerriam/nsdOtopy[114].

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

## Acknowledgements

This research was supported by the Intramural Research Program of the NIH (ZIAMH002966) to E.P.M. Collection of the NSD dataset was supported by NSF IIS-1822683 to K.K. and NSF IIS-1822929.

## Author contributions

Conceptualization, methodology, investigation, validation, writing—review and editing: Z.N.R., K.K., and E.P.M. Software, Data curation: Z.N.R. and E.P.M. Resources, supervision: K.K and E.P.M. Project administration, funding acquisition: E.P.M. Formal analysis, investigation, writing—original draft, visualization: Z.N.R.

## Funding

## Competing interests

The authors declare no competing interests.

## Additional information

**Correspondence and requests** for materials should be addressed to Zvi N. Roth.

