## [Peer Review File · Nature Communications]

Natural scene sampling reveals reliable coarse-scale orientation tuning in human V1REVIEWER COMMENTS

Reviewer #1 (Remarks to the Author):

While many studies have shown that fMRI voxels are tuned to orientation, there is currently some confusion in the field as to how this selectivity might arise. Some authors have pointed to stimulus confounds ('vignetting'), others have argued that voxel-based selectivity reflects an enhanced neural response for particular orientations. The current paper addresses this debate by showing that vignetting cannot explain voxel orientation selectivity. Rather, it appears that orientation maps obtained with fMRI reflect the responses of orientation-tuned neural populations.

I enjoyed reading this manuscript. It is clearly written, the work is technically solid, and the results are interesting. Given the widespread use of orientation stimuli and MVPA, these results will be of significant interest to the fMRI community. The paper could benefit, however, from more nuanced discussion of current and previous results, as outlined below.

- Tone down claims of novelty

While I really appreciate the model-based analysis approach, large data set and ecologically relevant stimuli to address the issue, the paper's conclusions are not entirely novel. For example, Carlson's group, who discovered vignetting (Carlson, 2014), also showed that voxel-based selectivity for orientation can be observed without vignetting (Wardle et al. 2017), much like the authors demonstrate here. This is not to say that I don't find the current results important or timely (they very much are), but I do think it would be appropriate for the authors to tone down their claims of novelty, and discuss and acknowledge these previous findings throughout the text.

- Some of the text is misleading

For example:

- Line 58-60: "However, this radial bias map, it turns out, does not reflect a physiological map.

Instead, it was shown to be a result of stimulus vignetting, an interaction between the edges of the stimulus aperture and the spatial frequency envelope of the stimulus (17, 18)."

- Line 334-336: "A number of prior BOLD fMRI studies have also reported a radial bias (13-16, 30). However, it has been recently shown that the radial bias reported in these earlier studies is due in large part, if not entirely, to a subtle stimulus confound arising from the influence of the stimulus aperture (17, 18)."

- Line 604-605: "We previously showed that orientation information measured by many fMRI studies is the result of stimulus vignetting (17)."

These claims are too strong, especially when read in context. While the maps of (13) were shown to be the result of vignetting in (17), no such definitive evidence exists for the maps observed in other work. Moreover, given the size of the stimuli used in the latter studies and Wardle et al. (2017), it seems very possible that at least of these previously reported effects were not driven by vignetting (see also below). Please revise.

- Line 169-172: "This result implies that in cases of low SNR, such as in conventional fMRI studies in which subjects are scanned at 3T across only one or a few scanning sessions, including orientation tuning in an encoding model will likely result in overfitting, and that doing so is unlikely to reveal reliable orientation selectivity. "

While I fully agree that SNR is important for detecting orientation selectivity, directly extrapolating to 3T seems a bit of a stretch. This is because SNR depends not only on voxel size and field strength, but also on the spatial scale of the underlying signals. For this reason, larger voxels at 3T might have somewhat comparable SNR to smaller voxels at 7T if orientation selectivity is represented in cortex at relatively coarse spatial scale (as reported here).

- Re-interpretation of previous results

The significance of the current work could be further emphasized by relating it more broadly to methods that capitalize on voxel orientation tuning, such as MVPA. More in-depth and nuanced discussion of previous findings on orientation maps would be helpful too. How do the current results affect the interpretation of previous fMRI findings on coarse-scale orientation maps (e.g. Sasaki et al. 2006, Furmanski & Engel 2000) and studies using MVPA? Do the current findings suggest that these studies may have tapped into neural orientation responses after all?

- Explanation of vignetting

It would be helpful for readers unfamiliar with the topic if the authors could explain/illustrate how edge effects might contribute to voxel orientation selectivity, and how their constrained model captures these effects in particular.

- Other relevant work that merits some discussion

Many others have contributed to this debate (e.g. Alink et al., 2013; Kamitani & Sawahata, 2010; Kriegeskorte et al., 2010; Pratte et al., 2016; Swisher et al., 2010), and acknowledgement of their contribution would be appropriate.

Reviewer #2 (Remarks to the Author):

This study tests for the coarse-scale orientation preference of radial bias in the human visual cortex while controlling for a possible confounding factor called stimulus vignetting. The study leverages a new large-scale fMRI dataset of participants viewing natural scenes, and a Gabor pyramid model is used to estimate and predict voxel responses to about 10k images. Initial attempts to fit the data do not indicate an advantage of the full Gabor model (which includes orientation as a parameter to be fitted) over the constrained model. However, if the authors select the more reliable voxels as identified from pRF mapping, a small but consistent advantage is observed for the full model over the constrained model. Given that the difference in model performance is so small (around 0.01), one wonders if the use of natural scenes data set is necessarily the best way to investigate orientation selectivity; the use of more tailored stimuli may be more effective. Additional analyses are performed, such as plotting the orientation preference of individual voxels and testing for deviations from other types of bias. The pattern of voxel preferences displayed in visual space are largely consistent with radial bias, and consistent with previous plots of orientation preference on the visual cortex as was shown by Sasaki et al. (2006).

While the technical aspects of the paper are quite strong, the case for novelty here is not clear. Most of the reported effects are consistent with earlier fMRI reports of the presence of radial bias (e.g., Sasaki et al., 2006; Freeman et al., 2011) and other biases (Freeman et al., 2013) in V1. There have also been reports of the presence of finer scale information in V1 (e.g., Yacoub et al., 2008; Swisher et al., 2010; Alink et al., 2017), which are discussed in brief in the current manuscript.

The assertion that stimulus vignetting might fully account for previous reports of radial bias seems to be raised in the manner of a straw man style argument, which the manuscript then provides evidence to refute. However, it seems very unlikely that all previous reports of radial bias might be attributable to stimulus vignetting. For example, there is anatomical evidence of elongation of the dendritic arbors of ganglion cells along a radial direction (e.g., Schall et al., 1986) which cannot be explained in terms of vignetting. The characterization of radial bias in single neurons with mapped receptive fields are also difficult to explain in terms of vignetting. Also, in the fMRI study by Sasaki et al. (2006) large field displays of gratings were used (~33 x 25 deg) and the gratings were of quite high spatial frequency (stripe thickness of 0.13 to 0.64 deg). This would greatly reduce any effects of vignetting; nevertheless, they observed effects of radial bias in V1 regions far from the edges of the stimulus.

Another limitation of this submission, as well as the earlier study of stimulus vignetting (Roth et al., 2018), is that not much information is provided about the Gabor pyramid model or the modeling results. In particular, the spatial frequency preferences of the Gabor units and the spatial frequency of the stimulus can have a strong impact on what pattern of vignetting is observed across 2D space. Attached is an example analysis that this reviewer performed on responses to vertical and horizontal sinewave gratings, summed across Gabor units tuned to different orientations and spatial phases at each location. Enhanced responses along the vertical axis of the annular grating are observed for Gabor units that match the spatial frequency of the stimulus as well as those that prefer a higher spatial frequency. However, units that prefer lower spatial frequencies actually exhibit the opposite pattern.

This analysis may partly explain why the study by Freeman, Merriam et al (2011) observed such strong apparent effects of radial bias in human V1 with low spatial frequency gratings (0.5 cycles per degree) and weaker effects of radial bias with higher spatial frequency gratings; the latter was also observed by Swisher et al. 2010. For low spatial frequency stimuli, stimulus vignetting will lead to responses consistent with radial bias because at near to mid eccentricities, V1 tends to prefer somewhat higher spatial frequency stimuli.

While the current study includes some interesting analyses and replications of previous work, the concerns of stimulus vignetting seem exaggerated and the main conclusions of the study are consistent with previous work that would not have been impacted by stimulus vignetting.

Reviewer #1:

While many studies have shown that fMRI voxels are tuned to orientation, there is currently some confusion in the field as to how this selectivity might arise. Some authors have pointed to stimulus confounds ('vignetting'), others have argued that voxel-based selectivity reflects an enhanced neural response for particular orientations. The current paper addresses this debate by showing that vignetting cannot explain voxel orientation selectivity. Rather, it appears that orientation maps obtained with fMRI reflect the responses of orientation-tuned neural populations.

I enjoyed reading this manuscript. It is clearly written, the work is technically solid, and the results are interesting. Given the widespread use of orientation stimuli and MVPA, these results will be of significant interest to the fMRI community. The paper could benefit, however, from more nuanced discussion of current and previous results, as outlined below.

We thank the reviewer for the positive assessment of our work.

- Tone down claims of novelty

While I really appreciate the model-based analysis approach, large data set and ecologically relevant stimuli to address the issue, the paper's conclusions are not entirely novel. For example, Carlson's group, who discovered vignetting (Carlson, 2014), also showed that voxel-based selectivity for orientation can be observed without vignetting (Wardle et al. 2017), much like the authors demonstrate here. This is not to say that I don't find the current results important or timely (they very much are), but I do think it would be appropriate for the authors to tone down their claims of novelty, and discuss and acknowledge these previous findings throughout the text.

We agree that Carlson (2014) discovered vignetting and we now clearly state this in both the Introduction (p. 3) and Discussion (p. 15). Our current findings differ from Wardle et al. (2017) in two important respects. First, Wardle et al. (2017) reported orientation information in the fMRI response, but did not demonstrate the source of that information (i.e., a fine-scale columnar bias or a coarse-scale map). We provide evidence for a coarse-scale map. Second, while Wardle et al. (2017) argued that the orientation information was not due to vignetting, we believe that the analyses demonstrated by the authors in their study do not satisfactorily rule out vignetting. Specifically, they found that voxels that respond to areas in between the stimulus edges enable orientation decoding, and this was interpreted as evidence for orientation selectivity that is not caused by vignetting. However, even voxels with a pRF center away from the stimulus edge may still respond to the edge, depending on the size (and shape) of the pRF. In other words, vignetting predicts that all voxels that respond to the stimulus will have orientation information, as long as their pRF does not drop to zero at the stimulus edge. In the current study, we developed a novel model-based approach that rigorously addresses vignetting. After we execute this approach, we furthermore show that there is indeed

substantial, coarse-scale orientation information in the form of a radial bias that cannot be attributable to vignetting.

- Some of the text is misleading

For example:

- Line 58-60: "However, this radial bias map, it turns out, does not reflect a physiological map. Instead, it was shown to be a result of stimulus vignetting, an interaction between the edges of the stimulus aperture and the spatial frequency envelope of the stimulus (17, 18)."
- Line 334-336: "A number of prior BOLD fMRI studies have also reported a radial bias (13-16, 30). However, it has been recently shown that the radial bias reported in these earlier studies is due in large part, if not entirely, to a subtle stimulus confound arising from the influence of the stimulus aperture (17, 18)."
- Line 604-605: "We previously showed that orientation information measured by many fMRI studies is the result of stimulus vignetting (17)."

These claims are too strong, especially when read in context. While the maps of (13) were shown to be the result of vignetting in (17), no such definitive evidence exists for the maps observed in other work. Moreover, given the size of the stimuli used in the latter studies and Wardle et al. (2017), it seems very possible that at least of these previously reported effects were not driven by vignetting (see also below). Please revise.

Here, the reviewer raises the possibility that some previous studies on orientation representation might have avoided the issue of vignetting. We feel the only definitive way to resolve this is to carefully re-analyze the previous studies, which would be out of the scope of the present paper. Therefore, to address this issue, we have contextualized and made more nuanced our stance (see p. 3, 15-16). In addition, we have added a new section to the Discussion (*Multiple scales of stimulus representation in humans*) in which we elaborate and explain our overall stance on past observations of orientation decoding.

Line 169-172: "This result implies that in cases of low SNR, such as in conventional fMRI studies in which subjects are scanned at 3T across only one or a few scanning sessions, including orientation tuning in an encoding model will likely result in overfitting, and that doing so is unlikely to reveal reliable orientation selectivity."

While I fully agree that SNR is important for detecting orientation selectivity, directly extrapolating to 3T seems a bit of a stretch. This is because SNR depends not only on voxel size and field strength, but also on the spatial scale of the underlying signals. For this reason, larger voxels at 3T might have somewhat comparable SNR to smaller voxels at 7T if orientation selectivity is represented in cortex at relatively coarse spatial scale (as reported here).

We agree with the reviewer's observations that there are a number of interacting factors that affect orientation information. Our intention in the text was simply to highlight how

the improved SNR of 7T and very large datasets, like NSD, can facilitate detecting subtle orientation information that is present at modest spatial resolutions. We have changed the text (Line 189-191) to reflect this.

- Re-interpretation of previous results

The significance of the current work could be further emphasized by relating it more broadly to methods that capitalize on voxel orientation tuning, such as MVPA. More in-depth and nuanced discussion of previous findings on orientation maps would be helpful too. How do the current results affect the interpretation of previous fMRI findings on coarse-scale orientation maps (e.g. Sasaki et al. 2006, Furmanski & Engel 2000) and studies using MVPA? Do the current findings suggest that these studies may have tapped into neural orientation responses after all?

Our view is that perhaps these studies tapped into neural orientation responses. But more likely, they were due to vignetting. Significantly complicating matters, an orientation signal not due to vignetting could arise from several distinct sources (a radial bias map, a cardinal bias, a columnar signal, etc) and these different sources of orientation signals could all coexist, but to varying degrees in different parts of the retinotopic map. The situation is considerably more complex than these earlier studies suggested. We now discuss the literature in greater depth than in the initial submission (see *Discussion: Multiple scales of stimulus representation in humans*), which hopefully clarifies the issues and strengthens the overall manuscript.

- Explanation of vignetting

It would be helpful for readers unfamiliar with the topic if the authors could explain/illustrate how edge effects might contribute to voxel orientation selectivity, and how their constrained model captures these effects in particular.

We have added a description of vignetting in the Introduction:

“The principle underlying vignetting is that the Fourier spectrum changes in the vicinity of a change in contrast, such as an edge. In standard orientation mapping experiments, the assumption is that each stimulus condition contains a single orientation. But because of the spread of Fourier power at the edge, this assumption isn’t true: different regions in the image contain different orientations and different Fourier power. Specifically, at the stimulus edge there is more power for the radial orientation than for other orientations.”

We have also added an explanation of how the constrained model captures vignetting effects:

“This model is image-computable and therefore accurately accommodates the stimulus edge and its associated consequences on orientation and Fourier power. The key feature of the constrained model is that equal contributions from all orientation-selective filters are enforced. Thus, if the constrained model is the model that most accurately describes voxel responses,

then we know that there is no voxel-level orientation selectivity. If, on the other hand, the full model is a best characterization of voxel responses, then we can conclude that there is indeed voxel-level orientation selectivity."

- Other relevant work that merits some discussion

Many others have contributed to this debate (e.g. Alink et al., 2013; Kamitani & Sawahata, 2010; Kriegeskorte et al., 2010; Pratte et al., 2016; Swisher et al., 2010), and acknowledgement of their contribution would be appropriate.

*We now cite all these studies in *Discussion: Multiple scales of stimulus representation in humans*. We thank for the reviewer for their insights.*

Reviewer #2 (Remarks to the Author):

This study tests for the coarse-scale orientation preference of radial bias in the human visual cortex while controlling for a possible confounding factor called stimulus vignetting. The study leverages a new large-scale fMRI dataset of participants viewing natural scenes, and a Gabor pyramid model is used to estimate and predict voxel responses to about 10k images. Initial attempts to fit the data do not indicate an advantage of the full Gabor model (which includes orientation as a parameter to be fitted) over the constrained model. However, if the authors select the more reliable voxels as identified from pRF mapping, a small but consistent advantage is observed for the full model over the constrained model. Given that the difference in model performance is so small (around 0.01), one wonders if the use of natural scenes data set is necessarily the best way to investigate orientation selectivity; the use of more tailored stimuli may be more effective. Additional analyses are performed, such as plotting the orientation preference of individual voxels and testing for deviations from other types of bias. The pattern of voxel preferences displayed in visual space are largely consistent with radial bias, and consistent with previous plots of orientation preference on the visual cortex as was shown by Sasaki et al. (2006).

The reviewer here mentions that natural scenes may not be as efficient as gratings to probe orientation selectivity. We agree, but note that this does not pose any intrinsic problems for the current study. We discuss this and related points in the revised manuscript (see *Discussion: Natural scenes vs. oriented gratings*).

While the technical aspects of the paper are quite strong, the case for novelty here is not clear. Most of the reported effects are consistent with earlier fMRI reports of the presence of radial bias (e.g., Sasaki et al., 2006; Freeman et al., 2011) and other biases (Freeman et al., 2013) in V1. There have also been reports of the presence of finer scale information in V1 (e.g., Yacoub et al., 2008; Swisher et al., 2010; Alink et al., 2017), which are discussed in brief in the current manuscript.

Here, the reviewer points out that the findings we demonstrate have some apparent history in the field. While this is true, we would like to point out the critical issue that despite the claims made by the previous studies, these studies did not provide sufficient evidence that support these claims. The contribution of the present work is to perform the requisite formal modeling, and after doing so, we find evidence of radial biases, and this is backed by a level of support that previous studies did not provide. Thus, even though the conclusions may seem reminiscent of past studies, the actual scientific advance here is new. To clarify these important points, we have revised the manuscript (see *Discussion: Multiple scales of stimulus representation in humans*). With this clarification, we hope that the reviewer agrees that the present manuscript provides valuable contributions to the field.

The assertion that stimulus vignetting might fully account for previous reports of radial bias seems to be raised in the manner of a straw man style argument, which the manuscript then provides evidence

to refute. However, it seems very unlikely that all previous reports of radial bias might be attributable to stimulus vignetting. For example, there is anatomical evidence of elongation of the dendritic arbors of ganglion cells along a radial direction (e.g., Schall et al., 1986) which cannot be explained in terms of vignetting. The characterization of radial bias in single neurons with mapped receptive fields are also difficult to explain in terms of vignetting. Also, in the fMRI study by Sasaki et al. (2006) large field displays of gratings were used (~33 x 25 deg) and the gratings were of quite high spatial frequency (stripe thickness of 0.13 to 0.64 deg). This would greatly reduce any effects of vignetting; nevertheless, they observed effects of radial bias in V1 regions far from the edges of the stimulus.

The reviewer is characterizing our position as being that all evidence of radial bias (anatomical, electrophysiological, imaging) is due to stimulus vignetting. We apologize that the initial manuscript gave this impression; we have now clarified in the revised manuscript that this is not our position. Rather, we believe that many prior fMRI studies on orientation were affected by stimulus vignetting. Without formal modeling, we feel it is impossible to definitively evaluate the extent of vignetting effects in prior studies. The value of our study is in developing an approach to distinguish between vignetting effects and true orientation selectivity.

For example, we believe that stimulus vignetting may account for the radial bias reported in Sasaki et al. (2006). Sasaki et al. (2006) did use a high-spatial frequency stimulus. But vignetting can occur for stimuli with either high or low spatial frequencies (Supplemental Fig S4). And although Sasaki et al. (2006) used a large stimulus aperture, they did not perform an eccentricity-based analysis in which they quantified the magnitude of the radial bias as a function of distance from the aperture edge. Moreover, in re-inspecting some of the maps from Sasaki et al. (2006), it appears that orientation selectivity is most robust at far eccentricities, in voxels with pRFs that likely overlapped the stimulus edge, and relatively weak or nonexistent at foveal and parafoveal eccentricities. Furthermore, as detailed above in response to Reviewer 1, vignetting cannot be bypassed through the use of large stimuli such as those used by Wardle et al. (2017) and Sasaki et al. (2006). While it is impossible to tell without a formal eccentricity-based analysis, Sasaki's results are not obviously inconsistent with stimulus vignetting. Regardless of whether Sasaki et al.'s results are indeed attributable to vignetting or not, the mere logical possibility is problematic, and highlights the need for explicit models and statistical model comparison, as we have performed here. We have revised the manuscript to be more explicit on these points (*Discussion: Multiple scales of stimulus representation in humans*), which we feel strengthens the case we lay out in the manuscript.

We agree that the series of papers from Schall and Leventhal are important because these papers provide a physiological (and neural) mechanism for the radial bias, and we discuss this in the Discussion section (*Discussion: Source of coarse-scale orientation bias*).

Another limitation of this submission, as well as the earlier study of stimulus vignetting (Roth et al., 2018), is that not much information is provided about the Gabor pyramid model or the modeling results. In particular, the spatial frequency preferences of the Gabor units and the spatial frequency of the stimulus can have a strong impact on what pattern of vignetting is observed across 2D space. Attached is an example analysis that this reviewer performed on responses to vertical and horizontal sinewave gratings, summed across Gabor units tuned to different orientations and spatial phases at each location. Enhanced responses along the vertical axis of the annular grating are observed for Gabor units that match the spatial frequency of the stimulus as well as those that prefer a higher spatial frequency. However, units that prefer lower spatial frequencies actually exhibit the opposite pattern.

The reviewer is correct that the effect of vignetting depends on the spatial frequency of the filter. In an orientation mapping experiment, grating stimuli are typically shown at a single SF, and therefore the most relevant filter of the pyramid is at that SF. Additionally, the stimulus SF should be around the preferred SF of the neurons in the region of cortex that is analyzed. In the current study we use natural scenes that encompass a wide range of SFs, and therefore we take all filters in the pyramid into consideration. In the Methods section we have now clearly stated the preferred spatial frequency of each filter. Preferred SF of the model filters were 21.85, 10.93, 5.46, 2.73, 1.37, 0.68, and 0.34 cycles/degree (128, 64, 32, 16, 8, 4, and 2 cycles per image). Bandwidth is 1 octave. We have also included an image of all filters in the pyramid in Supplemental Fig S3. Ideally, the filter responses across levels should be weighted differently for each voxel, reflecting that voxel's spatial frequency tuning (which, of course, would vary with eccentricity). Skipping this step, and instead summing across all filters will include spatial frequencies that do not activate the neural population within the voxel, and this may give an inaccurate characterization of the representations that the brain actually exhibits.

In Supplemental Fig S4 we demonstrate that the model filter that corresponds to the grating spatial frequency generally exhibits a radial bias, while the adjacent filters show an opposite effect. However, when scaling all filters to the same maximal value it becomes apparent that the effects in adjacent filters are much weaker than that of the filter with the maximal response. Therefore, as long as the spatial frequency drives the neurons in a given voxel strongly, and assuming a circular aperture, vignetting should result in a radial bias, across a wide range of spatial frequencies.

This analysis may partly explain why the study by Freeman, Merriam et al (2011) observed such strong apparent effects of radial bias in human V1 with low spatial frequency gratings (0.5 cycles per degree) and weaker effects of radial bias with higher spatial frequency gratings; the latter was also observed by Swisher et al. 2010. For low spatial frequency stimuli, stimulus vignetting will lead to responses consistent with radial bias because at near to mid eccentricities, V1 tends to prefer somewhat higher spatial frequency stimuli.

We agree with the Reviewer that the pattern of orientation bias produced by vignetting will depend on the spatial frequency of the grating stimulus, the shape of the aperture, and the location of the aperture edge in the retinotopic map, and all of these factors interact, sometimes in complex ways. We think the Reviewer is correct that the annulus aperture and the spatial frequency of the grating used by Freeman et al. (2011) probably produced strong vignetting effects. Note however that in Freeman et al. (2011), the radial bias for high frequency square-wave gratings was not really weaker than for low frequency sinusoidal gratings. What they observed looked like more of a vertical bias at mid-eccentricities. Perhaps the maps they reported reflect a conjunction of both stimulus vignetting and a true coarse-scale orientation bias.

As far as we can tell, Swisher et al. (2010) did not report the spatial frequency of the gratings they used. The stimuli they used did contain apertures with vertical 'pie' shaped cutouts that could have produced non-radial vignetting-related biases.

While the current study includes some interesting analyses and replications of previous work, the concerns of stimulus vignetting seem exaggerated and the main conclusions of the study are consistent with previous work that would not have been impacted by stimulus vignetting.

Here, the reviewer expresses concern about the novelty and impact of the present work.

First, the reviewer suggests that the manuscript's discussion of previous studies and vignetting seem exaggerated. We acknowledge that our discussion of past literature could use additional contextualization and increased balancing. We therefore have substantially revised the manuscript to provide this discussion (*Discussion: Multiple scales of stimulus representation in humans*). We do think that vignetting is still a major issue and the uncertainties related to vignetting cloud past studies. Hence, we feel that the methods and results we demonstrate in this paper provide a major contribution to the field.

Second, in addition to the scientific findings, we would like to emphasize the methodological value that the present work contributes to the fMRI field. There have been literally dozens of fMRI studies on orientation selectivity over the last nearly 20 years. We think part of the reason these studies have not reached consensus is because they have relied on ad hoc stimulus manipulations and ad hoc analyses. We believe this is a problem that extends far beyond visual neuroscience, and stems from the lack of formal models and rigorous model comparison (Merriam & Gardner, 2021). While we realize that the present work will not settle all debates regarding this issue, we do believe that our model-based approach represents a significant advance and provides a benchmark against which further studies in this area should be measured against. True, we end up supporting a claim made by others previously (i.e., that there is a radial bias, in the apparent absence of a columnar bias). But this should not cause one to overlook the importance of performing the modeling work that is necessary to provide

rigorous evidence for this claim. The revised manuscript now clarifies what we believe to be major contributions of the present study (see p. 15).

REVIEWER COMMENTS

Reviewer #1 (Remarks to the Author):

The authors did a very good job of addressing my previous comments. I have only a few remaining points that are easy to address:

Minor:

- Some of the text around Fig 4 & the caption is missing (I'm assuming that the text here should be identical to the previous version).
- Line 245-6: 'deviation from cardinal (Fig 5D, green)' is not shown in the figure.
- Fig 5, caption: 'G' and 'F' should be 'D' and 'E'.
- Supplemental Fig S4: typo in 'therefore'.

Signed by Janneke Jehee

Reviewer #2 (Remarks to the Author):

In this resubmission, the authors have revised the introduction and discussion sections of the manuscript to modify the previous assertion that stimulus vignetting may be entirely responsible for the presumed orientation responses observed in earlier studies, and here it is asserted that they are likely to be in large part responsible. The results and analyses remain the same and the discussion section has been modified to cover a broader range of the relevant literature.

The strengths of this study lie in its technical approach and the advanced computational analyses that the authors have applied to a large publicly available fMRI data set. However, the conceptual advances made by the study are not clearly or unambiguously articulated, and it is not widely accepted among researchers that stimulus vignetting can account for all or most of fMRI orientation decoding. The existence of coarse-scale radial bias has many diverse sources of positive evidence, and the current findings are in agreement with the conclusions of a sizeable body of literature. While not all of the previous literature is necessarily 100% bullet-proof, the weight of its full extent would be challenging to dismiss. Moreover, it is not clear why the authors claim in their rebuttal letter that the study by Sasaki et al. showed effects of radial bias that appeared more constrained to the stimulus edges, as the plots in Figures 3 and 4 of their Neuron paper, a prominent radial bias is observed that spans most of the visual cortex. (The specific foveal region is often not activated in studies because the fixation point is sizeable and modest fixational instability leads to inconsistent activation in this region in most studies, with activation emerging more parafoveally.) The broad extent of the radial bias found by Sasaki et al (2006) is conceptually consistent with the plots in Figure 3 of this paper, simply the analytic approach and the nature of the stimuli used to detect radial bias differ.

As noted, this study adds to an extensive body of work on fMRI studies of responses to visual orientation, with most of the recent studies having been published in more specialized neuroscience or neuroimaging journals. For these reasons, while the submitted work is deemed to be of high technical quality, its contents and broader conclusions are deemed better suited to be published in a journal that focuses on issues specific to vision and neuroscience research.

REVIEWERS' COMMENTS

Reviewer #1 (Remarks to the Author):

The authors did a very good job of addressing my previous comments. I have only a few remaining points that are easy to address:

We thank the Reviewer for the positive assessment of our work, and for the careful and constructive comments on both rounds of Review.

Minor:

- Some of the text around Fig 4 & the caption is missing (I'm assuming that the text here should be identical to the previous version).

This has been corrected.

- Line 245-6: 'deviation from cardinal (Fig 5D, green)' is not shown in the figure.

This has been corrected to '(Fig5A, green)'

- Fig 5, caption: 'G' and 'F' should be 'D' and 'E'.\

This error has now been corrected.

- Supplemental Fig S4: typo in 'therefore'.

This error has now been corrected.

Signed by Janneke Jehee

Reviewer #2 (Remarks to the Author):

In this resubmission, the authors have revised the introduction and discussion sections of the manuscript to modify the previous assertion that stimulus vignetting may be entirely responsible for the presumed orientation responses observed in earlier studies, and here it is asserted that they are likely to be in large part responsible. The results and analyses remain the same and the discussion section has been modified to cover a broader range of the relevant literature.

We thank the reviewer for their re-assessment of our manuscript. The reviewer is correct that the bulk of our revision was oriented towards expanding and improving the discussion of the significance and interpretation of our results, in line with the comments made by the two reviewers.

The strengths of this study lie in its technical approach and the advanced computational analyses that the authors have applied to a large publicly available fMRI data set. However, the conceptual advances made by the study are not clearly or unambiguously articulated [...]

Here, the reviewer disputes the conceptual advances that our manuscript makes. We respectfully disagree with the reviewer's assessment; we contend that our manuscript makes highly substantive contributions to the literature. The last decade has seen endless debate engendered by ad-hoc computational analyses (MVPA decoding analyses, RSA, and the like), which offer limited insight into the underlying neural computations. The most important conceptual advance of our manuscript is the development and application of neurally-inspired

image-computable models to characterize fMRI population responses, and the adoption of formal model comparison to adjudicate between competing theoretical accounts for the source of orientation-selective information. These observations are clearly articulated in the revised manuscript.

[...] and it is not widely accepted among researchers that stimulus vignetting can account for all or most of fMRI orientation decoding. The existence of coarse-scale radial bias has many diverse sources of positive evidence, and the current findings are in agreement with the conclusions of a sizeable body of literature. While not all of the previous literature is necessarily 100% bullet-proof, the weight of its full extent would be challenging to dismiss.

Here, the reviewer discusses the issue of whether stimulus vignetting can or cannot account for previous reports of orientation selectivity in fMRI measurements. Our view is that a careful assessment of past studies indicates that this issue is not actually known and is itself a worthy area of future research. Carlson's theoretical insights are convincing and suggest that the vast majority of prior fMRI studies could have been influenced by stimulus vignetting to some degree. Our previous work on stimulus vignetting (Roth et al., 2018) suggests that the orientation-selective signal can be strongly dominated by vignetting effects. Ultimately, these considerations highlight the unresolved confusion that the field has found itself in. In our manuscript, we provide a way forward by showing that it is possible to isolate and separately study the different sources of orientation selectivity using explicit models and formal model comparison. We have further clarified the stance of our manuscript with respect to coarse-scale radial bias claims (see p. 15, lines 464-468).

Moreover, it is not clear why the authors claim in their rebuttal letter that the study by Sasaki et al. showed effects of radial bias that appeared more constrained to the stimulus edges, as the plots in Figures 3 and 4 of their Neuron paper, a prominent radial bias is observed that spans most of the visual cortex. (The specific foveal region is often not activated in studies because the fixation point is sizeable and modest fixational instability leads to inconsistent activation in this region in most studies, with activation emerging more parafoveally.)

We disagree. In both Figures, there is a large signal drop-out at mid-eccentricities that extends all the way to the fovea. If this drop-out were indeed due to fixational instability, as the Reviewer suggests, then there should also be a similar signal drop-out in the maps of upper/lower visual field representations (Fig 5b) and the vertical/horizontal meridians (Fig 5c). But instead, those maps extend all the way to the fovea. While Sasaki et al. did not provide eccentricity maps, it is clear that the most strongly radially-selective voxels were located in the periphery, where pRFs are large and likely overlapped (to some degree) the edge of the stimulus aperture. But of course, Sasaki et al.'s results could reflect a combination of both stimulus vignetting and a true radial bias. Without model fitting, it is impossible to know. We discuss this issue on p. 15 (lines 440-444).

The broad extent of the radial bias found by Sasaki et al (2006) is conceptually consistent with the plots in Figure 3 of this paper, simply the analytic approach and the nature of the stimuli used to detect radial bias differ.

As noted, this study adds to an extensive body of work on fMRI studies of responses to visual orientation, with most of the recent studies having been published in more specialized neuroscience or neuroimaging journals. For these reasons, while the submitted work is deemed

to be of high technical quality, its contents and broader conclusions are deemed better suited to be published in a journal that focuses on issues specific to vision and neuroscience research.

We do not agree with the reviewer's assessment that the significance of the topics we study here are more suitable for specialized neuroscience/neuroimaging journals. We highlight that there are a number of high-profile papers pertaining to human orientation representation in journals such as *Neuron* (Sasaki et al.), *Nature Neuroscience* (Ling et al.), and *eLife* (Roth et al.). Beyond the results we obtain, our manuscript also emphasizes the generality of our computational approach to the broader neuroscience community, which has impact far beyond visual research.